# Linking genome size variation to population phenotypic variation within the rotifer, *Brachionus asplanchnoidis*

Claus-Peter Stelzer [1✉], Maria Pichler[1] & Anita Hatheuer[1]

Eukaryotic organisms usually contain much more genomic DNA than expected from their biological complexity. In explaining this pattern, selection-based hypotheses suggest that genome size evolves through selection acting on correlated life history traits, implicitly assuming the existence of phenotypic effects of (extra) genomic DNA that are independent of its information content. Here, we present conclusive evidence of such phenotypic effects within a well-mixed natural population that shows heritable variation in genome size. We found that genome size is positively correlated with body size, egg size, and embryonic development time in a population of the monogonont rotifer *Brachionus asplanchnoidis*. The effect on embryonic development time was mediated partly by an indirect effect (via egg size), and a direct effect, the latter indicating an increased replication cost of the larger amounts of DNA during mitosis. Our results suggest that selection-based change of genome size can operate in this population, provided it is strong enough to overcome drift or mutational change of genome size.

[1] University of Innsbruck, Mondseestr. 9, 5310 Mondsee, Austria. ✉email: claus-peter.stelzer@uibk.ac.at

Eukaryotic organisms display enormous variations in genome size, with haploid nuclear DNA contents ranging from 2.3 million base pairs in the microsporidian *Encephalitozoon intestinalis*[1] to 152 billion base pairs in the monocot plant *Paris japonica*[2]. Closely related species may differ substantially in their genome size, often by an order of magnitude[3–6]. Even intraspecific genome size variation has been described, among geographically isolated populations[3,7,8], among lab strains of model organisms[9,10], and in a few cases, within geographic populations[11,12]. On a mechanistic level, such genome size variations can be often attributed to variable proportions of repetitive DNA, in particular transposons[4,13–15]. Even though the number of genes scales positively with genome size across the domain Eukaryota[13], most species, and especially those with large genomes, carry much more genomic DNA in their nuclei than expected from their biological complexity and evolved functions[16]. Overall, the question as to why some eukaryotic genomes are streamlined, while others reach staggering sizes, still lacks a clear answer[13].

Current hypotheses on genome size evolution in Eukaryotes strongly differ in their emphasis on the evolutionary forces of mutation, selection, and drift. Theories focusing on mutations state that the genome size of a species represents a long-term equilibrium of mutations that increase and decrease genome size, by either referring to small indels ([17,18], but see[19]), or to the dynamics of transposable elements[20]. Variation among taxa is considered the result of biases in mutational rates, such that organisms with smaller genome sizes are able to remove DNA at faster rates than organisms with large genome size. In contrast, the mutational hazard hypothesis prioritizes drift as the main evolutionary force shaping genome size variation in eukaryotes[21,22]. It assumes a constant influx of mutations that increase genome size, which imposes a mutational hazard by increasing the genomic target size to deleterious mutations (in particular, harmful gain-of-function mutations). According to this hypothesis, variation in genome size mainly stems from differences in effective population size (hence, drift) among taxa. Large amounts of non-coding DNA may accumulate in small populations through drift, while at large population size this process is prevented by selection. Finally, selection-based hypotheses[23,24] emphasize that genome size could be indirectly selected through its correlations with various phenotypic traits. Selection-based hypotheses allow for scenarios in which nuclear DNA content is optimized. Even additions of non-coding DNA might sometimes be beneficial if they shift nuclear DNA content closer to an optimum level, while in other cases, genome streamlining might occur[25]. In analogy to the term "genotype", Bennett[26] coined the term "nucleotype", and nucleotypic effects, referring to those phenotypic effects of DNA that are independent of its information content.

Nucleotypic effects imply that genome size causally determines cell size and/or other life-history traits such as developmental rates and body size[26,27]. The main evidence in favor of this consists of ubiquitous correlations between genome size and different life-history traits in a variety of organisms[28,29]. However, correlation does not imply causation, and much of this evidence actually involves distantly related lineages, sometimes different genera, or even orders[30]. Such evolutionary units are probably separated by tens to hundreds of millions of years and can be expected to differ in many aspects other than genome size. Proponents of selection-based explanations cite polyploidy/chromatin diminution (i.e., programmed elimination of DNA during somatic cell divisions) as additional evidence for a causal link, because such mechanisms usually increase/decrease cell size within and among individuals[23,31]. However, opponents argue that such patterns could also be caused by dosage effects of genes

controlling cell size, and they point to an abundance of cell-cycle control genes that would render "nucleotypic control" of cell size unnecessary[21]. In summary, selection-based explanations of genome size are widespread and popular, but the evidence for nucleotypic effects still faces some caveats and limitations.

Some of these limitations can be overcome by studying evolutionarily recent changes in genome size, for instance, variation among subpopulations along geographic or altitudinal clines[32]. Nevertheless, gene flow is typically reduced between such populations, thus the above limitations mentioned still exist, even though to a lesser degree than in interspecific comparisons. To our knowledge, no study has yet convincingly shown that heritable within-population genome size variation significantly covaries with traits that are mechanistically associated with individual fitness (see, p. 34 in[21]). Such correlations, if they exist in a well-mixed natural population, would provide a much stronger case for nucleotypic effects, because confounding through different genetic backgrounds is avoided. Genetic variation affecting the phenotype should be randomized across different genome size classes (e.g., large, medium, or small genome size), thus avoiding biases in the genetic background among population members. Here we use the monogonont rotifer *Brachionus asplanchnoidis*, which allows to address such issues on a population level.

Monogonont rotifers are small metazoans, few hundred micrometers in size, found in fresh and brackish water habitats throughout the world. They have a facultatively parthenogenetic life cycle, which involves several generations of asexual reproduction, via ameiotic parthenogenesis, followed by occasional episodes of sex. In the genus *Brachionus*, sex is induced by proteins excreted upon population crowding, which trigger the production of mictic females and haploid males, which mate with each other and produce sexually recombined resting eggs. A rotifer clone refers to the asexual descendants of a single resting egg, thus each clone is a unique genotype. In lab cultures, it is possible to suppress sexual reproduction by frequent dilution intervals or large culture volumes, allowing the propagation of a rotifer clone for hundreds of asexual generations. Likewise, by inducing mixis in small culture volumes, males and mictic females from two different clones can be crossed with each other to produce outcrossed offspring, or mated within a clone, which is genetically equivalent to selfing.

In the present study, we focus on a population of *B. asplanchnoidis* from Obere Halbjochlacke (OHJ), a shallow alkaline lake in Eastern Austria. Recent studies have shown that this population harbors substantial and heritable within-population genome size variation[8,11]. Clones can be crossed with each other—even if they substantially differ in genome size. Genome size can be selected up or down by crossing individuals at the upper or lower end of the genome size distribution[11]. Genome size variation in the OHJ-population is mediated by relatively large genomic elements (several megabases in size), which segregate independently during meiosis and can thus be recombined to produce offspring that are variable in genome size. More recently, it has been demonstrated that these independently segregating elements consist of tandemly repeated satellite DNA, with only few interspersed genes or other sequences[33]. This strongly suggests that the extra DNA that is segregating in the OHJ-population has rather low information content.

In the present study, we took advantage of this natural genome size variation and investigate whether genome size correlates with a variety of phenotypic traits, such as body size, egg size, embryonic development time, asexual population growth, and the propensity for sexual reproduction. To this end, we analyzed body size and egg size variation in 141 genotypes of the OHJ-population, which were either sampled directly as resting eggs or

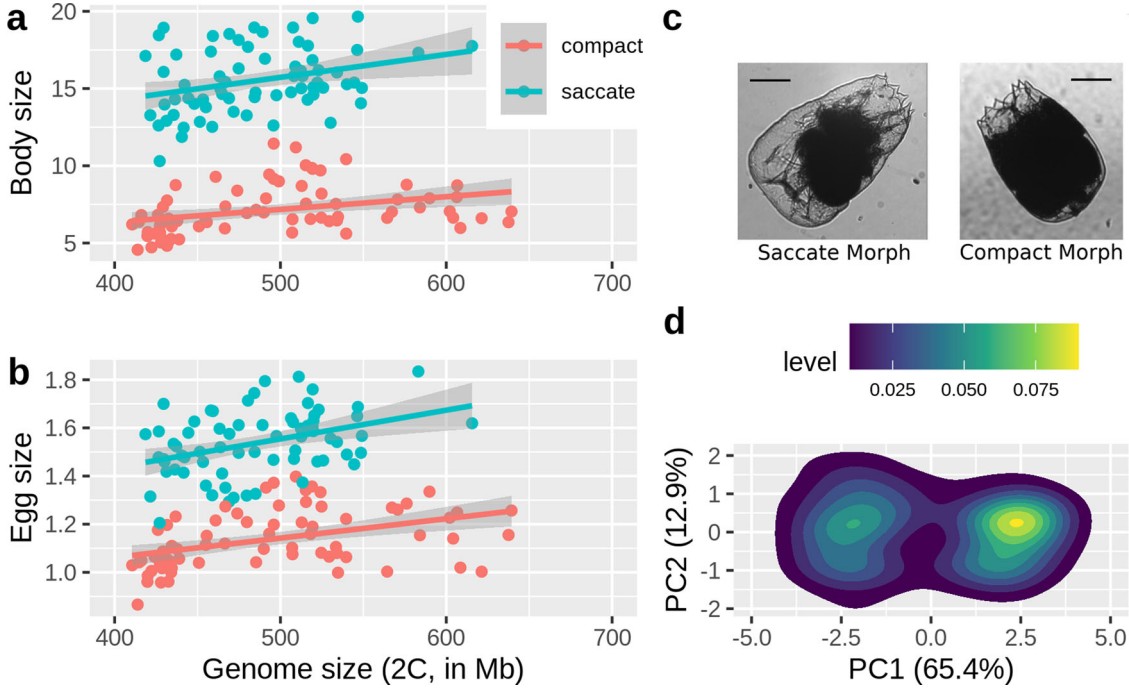

**Fig. 1 Relationship between genome size and morphotype on body and egg size. a** Correlation between genome size and body size (body volume in $10^6$ femtoliters) in 141 rotifer clones (c.f., Supplementary Data 2); **b** Correlation between genome size and egg size (egg volume in $10^6$ fl); **c** Representative photographs of the two morphs (Scale bars are 100 μm); **d** the two morphotypes could be distinguished by shape parameters using principle components analysis (here shown as a density plot, due to a large number of data points; see Supplementary Fig. 3 for display of individual clones and factor loadings). Source data of this figure are provided in Supplementary Data 1 and 2.

their outcrossed offspring. Genotypes were cultured clonally throughout this study and their genome sizes ranged from 414 to 639 Mb (mega bases, 2C value). In the same set of clones, we gathered data on asexual population growth and sexual propensity. In a subset of 17 clones, we additionally measured the embryonic development time. Our overall goal was to test whether within-population genome size variation correlates with any of these phenotypic traits.

## Results

**Size measurements**. Egg volumes and body volumes spanned a two- and four-fold range, respectively, from 0.8 to $1.8 \times 10^6$ femtoliters (fl) in the egg volumes, and $5–20 \times 10^6$ fl in the body volumes. This large variation was mostly caused by two discrete morphotypes present in the population, a larger "saccate" morph and a smaller "compact" morph. The discovery of these morphs was unexpected, but they could be distinguished by eye using light microscopy, or classified with principal components analysis, solely based on their body shape (Fig. 1, Supplementary Fig. 3, Supplementary Data 1, 2). Using the first principal component, we classified 69 clones in our dataset as saccate (threshold: PC1 < 0) and 72 clones as compact (PC1 > 0). Both morphotypes were found across the entire spectrum of genome sizes (Fig. 1). Within morphotype, all correlations between genome size and body size, or egg size, were significant (Body size/saccate: $r_{67} = 0.31$, $p = 0.011$; body size/compact: $r_{70} = 0.34$, $p = 0.0035$; egg size/saccate: $r_{67} = 0.38$, $p = 0.0012$; egg size/compact: $r_{70} = 0.43$, $p = 0.0001$). Egg size and body size were also significantly correlated with each other in both morphs (compact: $r_{70} = 0.82$, $p < 0.0001$, saccate: $r_{67} = 0.7$, $p < 0.0001$, Supplementary Fig. 4).

**Embryonic development time**. Morphotype did not significantly affect embryonic development time (EDT), but genome size did (Morphotype: $F_1 = 0.016$, $p = 0.899$; genomesize: $F_1 = 5.73$, $p =$

0.031). Combining the data of the two morphs resulted in a significant correlation between genome size and EDT ($r_{15} = 0.538$, $p = 0.026$, Fig. 2a). Since egg size was positively correlated with genome size (Fig. 1b), we used path analysis to distinguish between a direct effect of genome size on EDT and an indirect effect (i.e., an effect of genome size on EDT via egg size). In our dataset, the direct effect was about three times stronger than the indirect effect (Fig. 2c, Supplementary Table 1). A small proportion of eggs did not hatch during the experiment (1.9%, $n = 40$). These unhatched eggs were not different from viable (i.e., hatched) eggs in terms of their size (Supplementary Fig. 5). By contrast, the few occasionally picked male eggs (0.9%, $n = 18$), which are considerably smaller in volume, took almost 20% longer to hatch, compared to the mean EDTs of females of the same respective clone (Supplementary Fig. 5).

**Population growth and sexual propensity**. Finally, we checked whether genome size or morphotype affected sexual propensity (measured as the number of males produced by a clonal culture) and population growth rate (Fig. 3). While genome size did not affect sexual propensity, we observed a strong difference in between the two morphs, with the compact morph producing significantly more males than the saccate morph (Fig. 3a and b, Supplementary Table 2). In contrast, morphotype did not affect population growth up to a genome size of ~500 Mb, but population growth significantly decreased with genome size in the saccate morph (Fig. 3c and d, Supplementary Table 3).

## Discussion

In this study, we provided evidence that genome size in *B. asplanchnoidis* positively correlates with body size, egg size, and embryonic development time (EDT). To our knowledge, this is the first demonstration of such correlations in a well-mixed natural population with heritable variation in genome size. Earlier

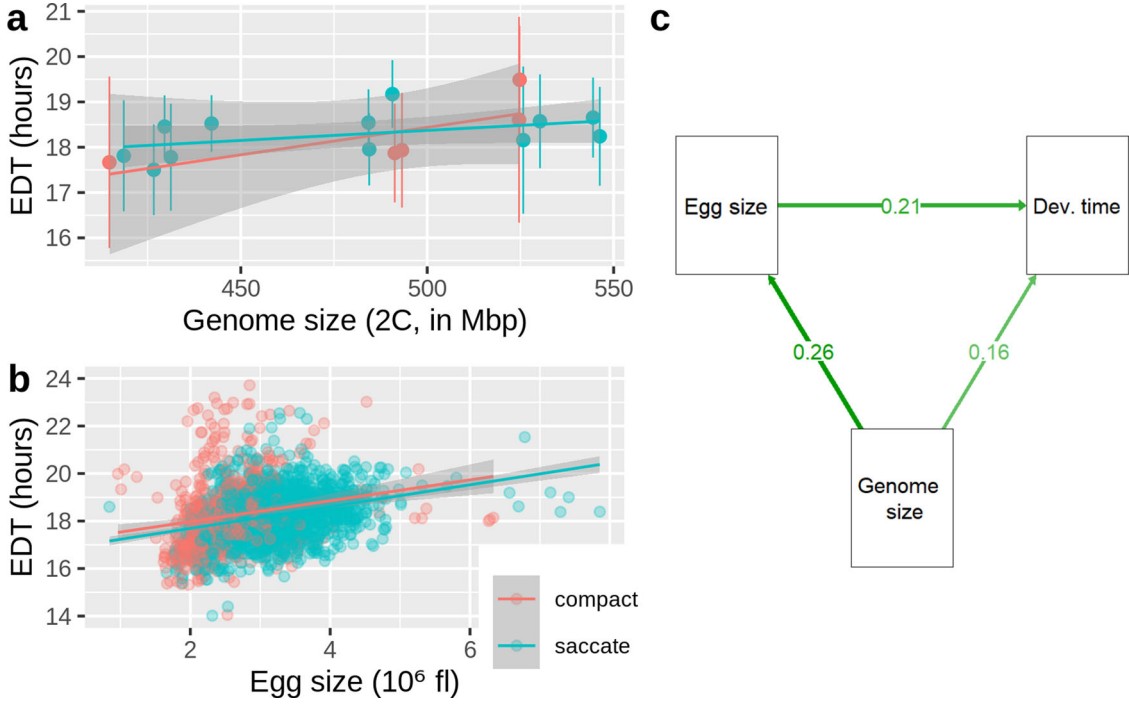

**Fig. 2 Relationship between genome size on embryonic development time (EDT). a** Genome size in 17 clones of the OHJ-population is positively correlated with average EDT. Shaded regions are the 95% confidence intervals. **b** Correlation between egg size and EDT, based on measurements of 2010 amictic eggs (Supplementary Data 3). **c** Path diagram showing direct and indirect effects (i.e., via egg size) of genome size on embryonic development time. The results of the underlying structural equation model are displayed in Supplementary Table 1. Source data of this figure are provided in Supplementary Data 3.

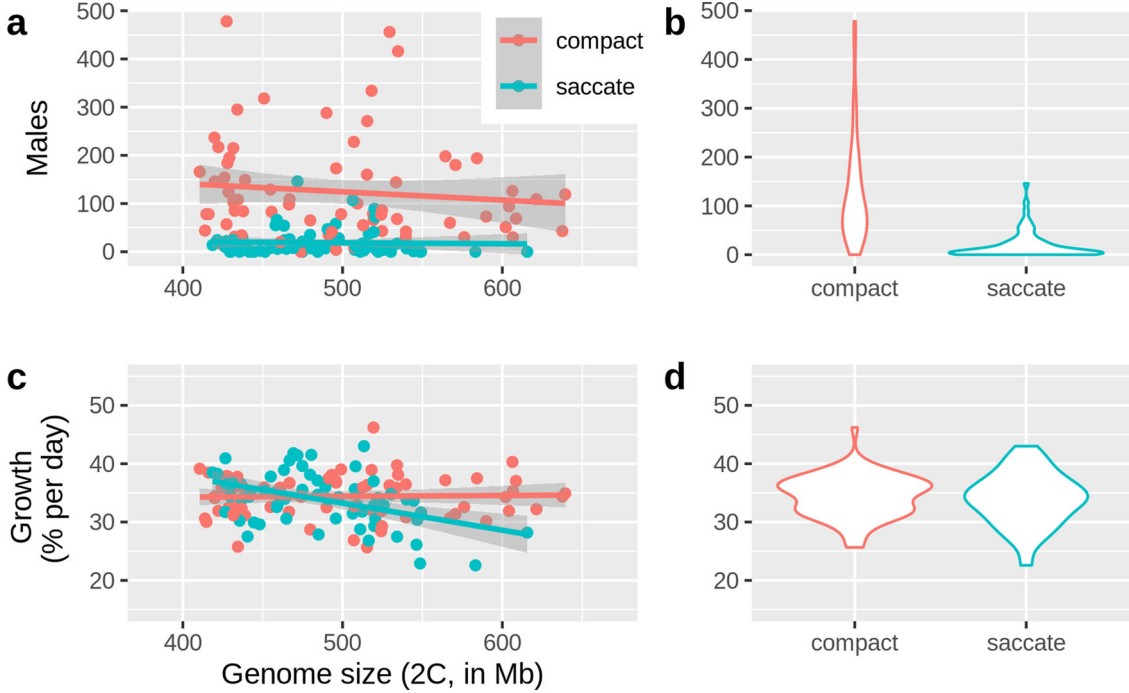

**Fig. 3 Relationships between genome size and morphotype on sexual propensity and population growth rate. a** Sexual propensity (number of males produced in dense clonal cultures) was not significantly correlated with genome size ($n = 141$ clones); **b** Morphotypes differed in sexual propensity, as the compact morph produced significantly more males than the saccate morph ($p < 0.001$, Supplementary Table 2); **c** Population growth rates were significantly reduced at higher genome sizes in the saccate morph ($p < 0.05$, Supplementary Table 3). **d** The two morphotypes did not differ in growth rate. Source data of this figure are provided in Supplementary Data 4.

studies on the OHJ-population confirm our assumption that it is genetically homogeneous, showing neither signs of genetic subdivision nor presence of cryptic species using molecular markers and barcoding genes[8,34]. Moreover, population members could be readily crossed with each other[8,11]. Population growth was also slower at larger genome sizes, but only in the saccate morph (Fig. 3c, Supplementary Table 3), possibly in part due to the slowed embryonic development of individuals with larger genomes. The fact that we only observed this effect when the large morph was combined with large genome size might indicate that strong deviations in body size are deleterious and would be selected against. However, these measurements have been done in a benign laboratory environment, and under natural conditions there could be situations where large body size is advantageous.

Nucleotypic effects on body size are particularly plausible for rotifers, which are eutelic and consist of ~1000 cells[35]. If cell size increases with genome size, body size should increase as well (unless there were fundamental constraints on maximum body size). Average cell size is difficult to quantify in rotifers since many tissues are syncytial in adults. However, egg volume might serve as a proxy for cell size, and thus allow comparisons among clones as asexual eggs are extruded before cleavage division. Consistently, egg size was correlated with genome size, and there was also a strong correlation between egg size and body size across our studied clones (Supplementary Fig. 4). In non-eutelic organisms, correlations between genome size and body size are expected to be weaker, since variation in cell number among individuals might blur the relationship between genome size and body size.

We also found that genome size correlates positively with EDT, another important life-history variable. EDT accounts for ~7% of the ~10-day life span in this species[36,37]. Two mechanistic pathways might be responsible for the prolonged duration of EDT at larger genome sizes, a direct effect, and an indirect effect via egg size. In many animals, EDT is positively correlated with egg size across a wide range of temperatures, indicating a universal physiological constraint[38]. Such an indirect effect via egg size was also present in *B. asplanchnoidis*, however, the direct effect was about three times stronger. Theoretically, this direct effect can be attributed to an increased replication cost of the larger amounts of DNA during mitosis[26]. In the future, detailed experiments on the embryological development of large vs. small genomes might clarify the exact cellular mechanism.

Genome size is certainly not the only factor determining body size in this population. In most animals, body size or weight shows significant heritability[39], so it is reasonable to assume that this trait might also be under polygenic control in *B. asplanchnoidis* as well. However, such genetic variation should be randomized across genome sizes in this well-mixed population. Sexual dimorphism is another source of phenotypic variation: rotifer males are typically much smaller than females, which is consistent with the fact that they are haploid. However, the longer EDT of male eggs (this study, and[40]) at first sight appears to contradict the relationship between genome size and EDT in female eggs, which contain twice as much DNA than male eggs. We suspect that this is due to additional processes/delays specific to male development. Within male eggs, we would expect that EDT increases with genome size. An examination of this possibility would require measurements of genome size in individual males, since male genome size can vary even within a clone due to meiotic segregation of the elements that cause genome size variation in this species[11].

Morphotype ("saccate" versus "compact") is an additional variable affecting body size in *B. asplanchnoidis*. We currently do not know the mechanism that causes these two morphs - it might be either genetic or epigenetic. Interestingly, non-genetic polymorphisms have been documented in several rotifer species[41].

They include discrete morphotypes, with variation in body appendages, presence/absence of anterior or posterior spines, or different body shapes. Such morphotypes often persist over many asexual generations, suggesting some form of epigenetic inheritance[41]. Morphotypes can additionally differ in physiological characteristics, for example sex-induction[41], which was also found in our study (Fig. 3b). Currently, we can only speculate about the adaptive significance of the two morphs. We suspect that they might be adaptations to different environmental conditions (e.g., different predators, salinity), but further studies are needed to establish this. Overall, the relationship between genome size and both body size and egg size was nearly the same in both morphs.

There are a few studies that have addressed genome size variation in populations and their phenotypic correlates, but with a slightly different approach. Huang et al.[10] examined genome size variation in the *Drosophila melanogaster* Genetic Reference Panel (DGRP), a collection of 205 inbred lines that derive from a natural population. Mated females were collected from the field and subjected to 20 generations of full-sib mating in the laboratory. Thus, the DGRP contains a representative sample of naturally segregating variation, but the high inbreeding coefficient of 0.986 predicts that most loci will be homozygous, including the many insertion (+) and deletion (−) alleles across the genome whose net number ultimately determines the genome size of an individual. Thus, we expect fixation of most (−) alleles in strains with the smallest genomes and fixation of most (+) alleles in strains with the largest genomes. Such individuals/genotypes are not very likely to occur in an outbred heterozygous natural population. In a follow up study, Ellis et al.[42] selected 50 lines representing the 25 of the largest and 25 smallest genomes from the DGRP, measured life history traits in all 50 of these lines. Thus, they only worked with the most extreme genome sizes. While this among-line genome size variation ultimately stems from structural variants that were already present in the original population, among-line variation in genome size is probably much higher than among-individual variation in a natural population. The DGRP approach offers greater resolution of genome size—phenotype connections than is possible with studies of wild individuals, and indeed this study uncovered some interesting phenotypic correlates[42]. However, applying these genome size—phenotype relationships to the original population is not straight forward due to the inbred history. In our study, clones represent the natural genotypes and nucleotypes, with essentially the same level of heterozygosity as in the field population. Thus, a correlation between genome size and a phenotypic trait can be directly applied to the population, especially since genome size is inherited like a quantitative trait with a heritability of 1[11].

Another recent study on flies examined whether long-term phenotype selection results in genome size differences between lines that experienced contrasting selection regimes[43]. Lines selected for fast/slow development did not change in genome size as expected (i.e., decreasing/increasing in genome size), instead they all converged to the same mean genome size, but with reduced genome size variation compared to the founding populations. Likewise, selection for body size did not result in significant differences among lines in outbred selection lines maintained at large effective population size. However, isolines derived from such populations after ten generations of full-sib mating experienced greatly increased genome size variation and "bloating" of genome size in several individuals. This observation closely resembles the pattern of increased genome size variation in the DGRP lines (discussed above), and it matches our own observations in *B. asplanchnoidis*, where repeated selfing can lead to surprisingly high genome size variation and to increases in genome size compared to the founding individual (see Fig. 3 in[11]).

Our tentative conclusion is that patterns in genome size derived from highly inbred lines might not be representative for genome size evolution in a natural population. Even though such observations are highly interesting and potentially informative about the basic mechanisms of genome size variation, we decided to not include selfed lines in our present study.

Finally, one might argue that there are many studies on B-chromosomes and their correlations with phenotypic traits (e.g.,[44–46]), which might also qualify as evidence for a genome size—phenotype relationship. B-chromosomes are one possible manifestation of intraspecific genome size variation, in which extra DNA is organized into accessory chromosomes rather than as insertions/deletions to the normal chromosomes. Thus, it may be tempting to consider the number of B-chromosomes in those studies as a proxy for genome size and to interpret these correlations accordingly. However, B-chromosome numbers can be very misleading. For example, in maize extra DNA can be present as B-chromosomes and/or chromosomal knobs, which are large heterochromatic regions on normal chromosomes. Intraspecific genome size variation between cultivars or across altitudinal clines is mainly mediated by chromosomal knob content[32,47], yet surprisingly, Bs and knob content are often inversely correlated[48]. Thus, while studies on B-chromosome variation are certainly interesting in their own right, simple extrapolations from the number of Bs towards genome size are not appropriate. To make inferences about the effects of extra DNA, there seems to be no way around directly measuring genome size.

In this study, we provide conclusive evidence that additional DNA, which is present in the genome of some individuals of a population but missing in others, exhibits nucleotypic effects. In *B. asplanchnoidis*, this additional DNA is largely non-coding, mainly consisting of tandem repeats of satellite DNA with only few interspersed genes[33]. This suggests that the studied population meets the basic requirements for genome size to be selected up or down, indirectly, through its correlations with several important life-history traits. So far, model organisms allowing such a population genetic perspective have been missing in the literature. In this regard, Brachionid rotifers are very suitable for even experimental approaches, owing to their short generation times and ease of culture[49,50]. In addition, genomic tools are becoming increasingly available for these species[15,51–53]. Future investigations in this system could address selection-based mechanisms in more detail experimentally, and allow to quantify their relative importance and interactions with mutational- or drift-based mechanisms, in order to explain genome size evolution over (micro-) evolutionary time spans.

## Methods

**Origin of clones and culture conditions**. Resting eggs of rotifers were collected in autumn 2011 from the sediments of OHJ, a small alkaline playa lake in Eastern Austria (N 47°47′11″, E 16°50′31″). Animals were kept as clones, which consist of the asexual descendants of an individual female that originally hatched from a single resting egg. Since resting eggs are produced sexually in Monogonont rotifers[35], each clone is a unique genotype. All clones have been characterized previously with regard to their genome size, and we use the same nomenclature as in this earlier publication[11].

Rotifers were cultured in F/2 medium[54] at 16 ppt salinity and with *Tetraselmis suecica* algae as food source (500–1000 cells μl$^{-1}$). Continuous illumination was provided with daylight LED lamps (SunStrip, Econlux) at 30–40 μmol quanta m$^{-2}$ s$^{-1}$ for rotifers, and 200 μmol quanta m$^{-2}$ s$^{-1}$ for algae. Stock cultures were kept either at 18 °C, re-inoculated once per week by transferring 20 asexual females to 20 ml fresh culture medium, or they were kept for long-term storage at 9 °C, replacing approximately 80% of the medium with fresh food suspension every 4 weeks.

**Size measurements**. All body and egg size measurements took place soon after a clonal culture had been established, usually within 1–2 months after hatching from the resting egg. The entire measurement campaign lasted for three years and accompanied measurements on genome size variation in this population[11]. In total,

we analyzed body size and egg sizes of 141 different *B. asplanchnoidis* clones. Some of our rotifer clones directly hatched from the natural population, while others were the sexual (outcrossed) offspring of the founders in the F1 or F2 generation[11]. Three independent preparations on different dates were made for each clone, which consisted of collecting ~20 adult females with two attached asexual eggs from stock cultures, exactly one week after re-inoculation and growth at 18 °C. Rotifers were fixed in Lugol's solution, and eggs were detached from the females by vigorously vortexing the fixed sample for three minutes. Eggs and females were then collectively transferred into a well of a 96-well plate (Greiner, no. 655096). Each well was carefully topped off with culture medium and covered with a microscope coverslip. Body sizes and egg sizes were determined using a custom-made digital image analysis system, which used custom algorithms written in the NI Labview and NI Vision software packages (National Instruments) for controlling hardware, image acquisition, and image processing (for details, see Supplementary Methods and Supplementary Figs. 1, 2). Body and egg volumes were calculated from length and width measurements by approximating the shape of an ellipsoid (main axis = length).

**Embryonic development time**. Embryonic development times were measured in a subset of 17 rotifer clones, which were selected to represent the genome size range of the natural OHJ-population (404–552 Mb). We quantified the duration of embryonic development, i.e., the time from egg extrusion until hatching, using a time-lapse recording system, which was a modified version of our image analysis system mentioned above (see Supplementary Methods and[55]). With this setup, we could simultaneously follow the hatching phenology of up to 96 embryos over the course of one day. Production of eggs, as well as checks for hatching, were confined to 30 min-intervals. For each of the 17 rotifer clones, the hatching times of 96–141 asexual eggs were recorded.

**Population growth and sexual propensity**. We also obtained estimates of clonal population growth and of the propensity of clones to induce sex. This was also done along our earlier measurement campaign of genome size variation while growing biomass for flow-cytometric measurements[11]. Briefly, 1-L culture flasks were inoculated with 60–300 females (depending on preliminary screens of population growth in the stock cultures), with each female carrying two asexual eggs, and they were allowed to grow for seven days. Three replicates per clone were used. After 1 week, two 1 ml samples were taken from the well-mixed culture, and females and male rotifers were counted. Population growth was calculated as (ln $N_t$ − ln$N_0$)/t, where $N_0$ and $N_t$ represent the number of females at inoculation and after 1 week, respectively. The number of males in the three $1+1$ml-samples served as a measure of sexual propensity.

**Statistics and reproducibility**. All statistical analyses were done in R (v4.0.2,[56]). During initial data exploration, it became obvious that the OHJ-population consisted of two discrete morphotypes, which differed in size and shape (a larger "saccate" and a smaller "compact" morph). To objectively identify these two morphs, we used principle components analysis on rotifer body shape with the function *prcomp*. For graphical visualization we used the package ggfortify[57] and its function *autoplot*. The input variables for PCA consisted of eight different (size-independent) shape parameters, which we obtained through automated particle measurements implemented in the NI Vision software package (for more details, see Supplementary Methods). For example, the so-called compactness factor relates the area of a particle (i.e., the binarized picture of a rotifer) to the area of its bounding rectangle. We calculated PCAs for two levels of data aggregation, on clone means ($n = 141$) and on measurements of each individual female ($n = 5644$). Pearson correlation tests of genome size *vs.* body size, egg size, and embryonic development time were done in the standard module of R with the function *cor.test*. To disentangle a direct effect of genome size on embryonic development time from an indirect effect (i.e., via egg size), we carried out path analysis using the R-package lavaan[58] and semPlot[59]. The effects of morphotype or genome size on sexual propensity (=number of males in dense populations) and population growth were examined using generalized linear models, using the functions *glm.nb* (MASS package,[60]) and *glm* with negative binomial and quasibinomial error structures, respectively.

**Reporting summary**. Further information on research design is available in the Nature Research Reporting Summary linked to this article.

## Data availability

The data supporting the findings of this study are available within the article and their Supplementary Data files. The source data underlying Figs. 1–3 are provided in Supplementary Data 1-4. All other data are available from the corresponding author upon reasonable request.

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

## Acknowledgements

We thank Peter Stadler for help during routine maintenance of the rotifers, and Simone Riss for providing helpful comments on the manuscript. Alois Herzig provided access to the sampling site and helped during sampling of resting eggs at Obere Halbjochlacke. Funding for this project was provided by the Austrian Science Fund, grant number P26256 to CPS.

## Author contributions

C.P.S. conceived the study, designed, and participated in all experiments. M.P. contributed the size measurements and measurements of population growth and sexual propensity. A.H. contributed the embryonic development time measurements. A.H. and M.P. did most of the culturing work. C.P.S. analyzed the data and drafted the article. All authors read and approved the final article.

## Competing interests

The authors declare no competing interests.
