## [Peer Review File · Communications Biology]

Reviewers' comments:

Reviewer #1 (Remarks to the Author):

In the manuscript "Evidence for nucleotypic effects on the level of a population" the authors take advantage of genome size variation within a population of the rotifer *Brachionus asplanchnoidis* to investigate the effects of genome size on phenotypic correlates, such as cells size and development time. Overall, I really enjoyed reading this paper. I feel it was well thought out and designed and really makes me excited to see what work comes from this. I feel it is one of the few papers to really directly investigate the question they are asking rather than go about it in a roundabout way. I am certainly excited to see this work published, as it definitely will be a good reference for the field of genome size evolution, especially concerning within species genome size change.

I have a few larger comments/questions I feel should be addressed, but most suggestions I have are very minor. Mostly, I wonder how the authors are defining "within population studies", as some work that is published is within isolines developed from single collection events. How does this differ from what they consider "within population". Additionally, at least one work was recently published on phenotypic correlates with genome size (although a very different experimental design and less clear results). I wonder what the authors think about how that work relates to their work, or if it does not. There is one area where I hope to get some clarification on statistical methods (Figure 3c). I am not sure if an interaction term was tried? Is there a statistical difference in slopes for morphs when looking at GS vs growth rate.

Major:

Lines 84-85: This is where I have my biggest issues/questions about your introduction and discussion. This is an interesting question for sure (impact of genome size on phenotypes). Certainly, much work has been done on among population differences in genome size. Within a population can be a little trickier, however. How do you consider variation within isolines developed from single populations. For example, the DGRP (Huang et al .2014) have 205 lines of *Drosophila melanogaster* which originated from a single collection event in Raleigh. These isolines could then breed for generations, then be sequenced. There was a large difference in genome size (ranging from 165-190 Mbp or so). Although, phenotypes were not correlated in initially. But a follow-up study (Ellis et al 2014, Plos Genetics) looks at reproductive fitness/phenotypes related to temperature in lines with large and small genome size. Additionally, another recent study published in Genes (Hjelmen et al. 2020) looked at genome size in flies selected for generation time (blow fly) and flies selected for body size (*drosophila melanogaster*). There was variation in either instance, but not clear evidence there was directional change in genome size. Selection for a larger body size did increase variation in genome size, would suggests a reduction in constraint. This study was very different and used pre-exciting selection lines (selected for studies of phenotype) and later measured genome size. This may be something worth mentioning or discussing/thinking about.

Line 124: I may have missed it here, but are these slopes significantly different (between morphs)

Figure 1b: I would suggest moving this legend slightly, as it appears to be covering at least one point slightly. Not sure if any are missing behind it. (not likely)

Figure 2a & b: can you specify what the shading indicates? Is this a 95% confidence interval?

Figure 2c & d: font size on C and D are far too small to be legible. Can you increase sizes here?

Line 156: What test statistic here for significance? Supplemental data suggests that significant at $p = 0.048$, when combining all data. At least with morph as a component in your model. What does it look like if you don't include morph in your model or if you use an interaction term? It seems that the morph may be driving some. The saccate morph has a negative slope, but the compact does not have a convincing negative slope. Have you run the model with an interaction effect between morph and GS?

Line 170: Again, i am interested to see how much of an effect the morph has, it does really seem different. This could easily be a trick of the eye, though.

Line 172: Again, how would you differ this from the DGRP results in Ellis et al 2014 with temperature/gs reproductive fitness phenotypes? This study had pupation time, pupal mass, eclosion time, etc.

Line 210-211: I do think this would be interesting to see. As referenced in my comments earlier, there does seem to be an effect of genome size and plasticity in development for *Drosophila melanogaster* lines

Minor: I would suggest putting the species name of the organism into the abstract, rather than just "small aquatic metazoan"

Line 22: I'd be careful with a word like "necessary", what about cases for junk dna?

Line 52 (references to mutational equilibrium): Some may argue you should include statements that this mutational equilibrium model likely couldn't account for the large differences we see (it's too slow) (Gregory 2004, Insertion--Deletion biases and the evolution of genome size). Another reference that is more recent covering the same idea but at a faster/larger scale would be the "accordion" model in Kapusta et al. 2017. As is, I find this to be a nice summary, though.

Line 71: can you give a reference for the statement about "sometimes even genera within families"

Line 114: can you spell out "fl" the first time you use it here.

Reviewer #2 (Remarks to the Author):

This paper poses an interesting general question about variation in genome size, but then the study itself is a bit more specific on interesting biology related to rotifers. The data addresses variation in genome size but also encompasses phenotypically plastic body size morphotypes. My main comment is that the general aspects and justification do not fit well with the specific findings, and so I would recommend that the background to the study is reformulated to fit more closely with the study system and the actual data presented here, which are good.

The main limitations are:

1) It is unclear what the elements are that contribute to variation in genome size. The implication in referring to the nucleotype hypothesis is that it is non-coding, but that evidence is not presented. If it includes any genes, then this could change interpretation of the consequences of the extra material. Then there would be an alternative to the nucleotype hypothesis – that there is a dosage effect of some genes that influence body size – cannot be ruled out by the present data. Hence, results do not support the title. Also, is there any evidence whether they display selfish behaviour or biased transmission?

2) Claims that this is the first study seem to miss lots of work on accessory B chromosomes. I googled "B chromosomes and body size" and found several papers in mammals and plants reporting either positive or negative correlations of body size with number of B chromosomes, and hence genome size. This work should be referred to and the context for the present study refined accordingly.

3) The introduction contains a lot of information about variation in genome size in general, building up to this as a test of the general idea for variation in genome size among eukaryotes. But the discussion does not return to these general questions convincingly – how do the specific results of this study answer the big questions asked at the start? Is this variation really relevant for some of

those questions, how does it relate to previous cases of shifts in ploidy or accessory chromosomes or TE contents? Perhaps best would be to rework the beginning and focus more closely on the things of interest here that occupy more time in the discussion, i.e. the rotifer context and possible consequences of variation for them.

4) The word nucleotype is not widely used or known as far as I can tell, and might be better replaced in abstract/title to avoid unnecessary jargon. Neither the title nor abstract mention the study organism - for a single-species experimental study of this kind with lots of biological detail (e.g. morphotypes) should really name the study organism in the title and abstract.

Line 51. The distinction between your two mutational hypotheses is not entirely clear as worded. Both of these rely on mutation and drift?

Line 73. What is chromatin diminution?

Line 88. Are you sure that the different genome sizes do not include gene presence/absence or copy number, which could still permit the dosage explanation?

Line 96. It would be useful to know more about these elements – are they accessory chromosomes? Do they contain any genes or purely non-coding DNA?

What explains the saccate and compact morphs? This appears as a complication in your study, which is fine and needs to be addressed, but would help reader to have a bit more explanation of what these are. E.g. are they something like defence polyphenism in *Daphnia*, which people might know about?

Do the data in figure 1 include artificially selected lines as well or just the natural ones?

Line 137. Not fully explained what direct and indirect effect you are talking about – please clarify what factors these effects are between and via in each case.

Line 140-143. Losing focus here, why are you interested in the male eggs and their hatching time? Not clear to the reader. Could this be tied to differences in genome size as well if males are haploid?

Line 156. "Population growth rate declined significantly with genome size" – from figure 3 it looks like this is just for the saccate morph. Previous parts separated effect between the two morphs it is unclear why you lumped everything together for this part.

Line 225. Please can you add the sampling date.

With permission of the editor, my research group helped to review this manuscript, but the final report is my own.

Reviewer #3 (Remarks to the Author):

This manuscript details a study that identifies the existence of nucleotypic effects within a genetically homogeneous population. The authors generally conclude that the body size, egg size, and embryo development time can each contribute to the genome size. While this study demonstrates some new approaches, I possess several significant reservations regarding the study that preclude its publication in *Commun. Biol.* below. Some points of concern with the study are outlined below, alongside some more specific line-by-line comments. Given my concerns, I unfortunately must recommend rejection of the manuscript in its present form.

Major comments

(1) Novelty: It is not clear what the novelty of this study is. Although the authors said that the two morphotype (saccate and compact) outweigh the nuclear effects, they have not presented the relevant mechanisms. The abstract describes evidence for additional DNA and nucleotype effects,

but the authors only provide correlations between the genome and egg size, embryo development time and phenotypes. Furthermore, two other studies are given (refs 24 and 25) that already looked at the correlation between genomes size and life history traits.

Additionally, there is already substantial evidence of a weak relationship between genome size variations and organic complexity (C-value enigma). Indeed, the crucial considerations of this paradox were mentioned in the Tomas 1971 review and in Nature Ecology & Evolution by Liedtke. Therefore, it is unclear how this current research has added any novel or improved insight.

(2) The Introduction needs more information on the species' biology. If, as the authors said, resting eggs from each clone are unique genotypes, why would you expect correlation between genetic variation and life history in cyclic parthenogenesis? (The authors cited the need to avoid biases in the genetic backgrounds among population samples) (L89-90). Therefore, I would argue that cyclic parthenogenesis is not appropriate because too many variables must be taken into account to compare genomic variation with life history (i.e., mutation, meiosis).

(3) Unfortunately, the authors did not examine the single nucleotide polymorphism to estimate genotypic diversity among their samples in population-level analysis. In addition, the morphotype is more significant than the nucleotypic effect (L197-198), but it is not clear because it is not specifically mentioned. Without a fully-factorial design, the authors cannot really distinguish the correlation between genome size and life history characteristics, and this precludes many of their conclusions.

(4) The description of the Methods requires more detail in order for the reader to gauge the methodological approach. There were many instances where I could not understand what the authors were trying to say. I had a hard time understanding the experimental procedure, and am still not sure what was done.

(5) The Discussion section lacks of an integrative analysis of its three components (nucleotypic effect, life history traits, and morphotype). In Result, the authors stated that the correlation between genome size, morphotype, and life history traits, but in-depth analysis of their relationship in Discussion was absent. At least, an effort should be made in this direction in the Discussion of results. Thus, this section should be rewritten, as it lacks depth.

(6) Further, despite using statistical figures such as PCA and a generalized linear model based calculation, there are no details provided in this manuscript for how the calculation was performed or of the tool that was implemented. Therefore, the statistics section needs to provide substantially more detail as, in this current form, there is not enough information from the statistical design to adequately judge the statistical approach.

Specific comments

L34: The abstract should include a brief statement explaining the primary results and a short concluding sentence. The section on 'selection' should be omitted as this part of the study was not investigated.

L83-87: This is not clear to me. In this experiment, isn't an asexual rotifer more suitable to avoid total disruption due to genetic differences? (Line 241: you used different clones in this experiment).

L115-116: What are the criteria for selecting the two types? Many studies have recognized three major morphotypes: large (L), small-medium (SM), and small (SS). (Fu et al., 1991).

L171-172: What if different populations had different trends?

L175-179: I cannot verify the validity of this statement based on the data presented in the figures. Please point the reader to the data identifying the inheritance of body size in individual clones.

L192-193: Include a reference for the statement that non-genetic polymorphisms can occur in several rotifer species.

L199-201: I am confused by this section. Does it mean that the nucleotypic effect is more important than the morphotype because the selection works well on all the phenotypes or the phenotypes are sometimes observed in only one form? This statement is perplexing.

Reviewer #4 (Remarks to the Author):

The authors take advantage of a species, the monogonont rotifer *Brachionus asplanchnoides*, that has reproductively compatible isolates of drastically different genome size to look for "nucleotypic effects" – phenotypic effects of genome size independent of genetic information. They convincingly demonstrate a positive correlation between genome size and female adult body size, egg size, and

embryo development time. This is the most direct evidence yet of nucleotypic effect, free of the confounding variables of evolutionary and ecological divergence that comes with cross-species comparisons, and suggests that *B. asplanchnoides* would make a suitable system to study whether these phenotypes are under selection. The paper is well written and admirably concise, and I recommend that it be accepted after minor revision.

The authors definition of nucleotypic effects seems to assume that the "extra" DNA segregating in the population is not transcriptionally active ("non-coding" "independent of information content"). It is probably full of transposons, and if they have recently been introduced into a new genomic environment, they may be quite active and conferring their own deleterious effects. If the authors could say anything about what is known about the nature of the excess DNA it would be helpful.

The authors describe two morphotypes, one considerably larger than the other. This is an interesting thing in and of itself for a different publication, but the fact that the two sizes have nearly the same relationship between genome size and both body size and egg size is a powerful indicator of how these nucleotypic effects scale. It would be interesting if these relationships held with males. Fig S5 briefly touches on development time of male embryos—male embryos develop faster than females from the same clone—but a more interesting question is whether males show the same relationships as females in Figs 1 and 2.

I don't understand Figure 3c. The authors state that population growth rates were significantly reduced at higher genome sizes and present the results of a GLM in Supplemental, but did this model lump compact and saccate morphs? Eyeballing Fig3c the two seem to be different, with saccate decreasing with genome size and compact remaining constant. If population growth rates are really reduced with increasing genome size this would seem to be a serious deleterious effect.

In the Abstract, the authors state that the observed nucleotypic effects "account for" difference in genome size. This seems to conflate two ideas: one, that changes in genome size have non-genic phenotypes, and two, that these phenotypes can be subject to selection in a meaningful way. The results support the first idea but as the authors make clear in the discussion more work is needed to establish the second. Perhaps something like "Nucleotypic effects are predicted to arise from variation in genome size—they refer to..." would be better.

In a similar vein, the organization of the second paragraph of the introduction implies that nucleotypic effects are an aspect of selection-based hypotheses for variation in genome size. I'm not sure this has to be true, or that the "mutation hazard" and "selection based" hypotheses are mutually exclusive. The first assumes excess DNA is mildly deleterious and the second assumes it can be mildly beneficial or mildly deleterious. The first emphasizes the importance of drift vs selection in small populations while the second assumes extremely efficient selection over a short period of time. The question is whether the selective effects are strong enough (or populations large enough) to overcome drift, with the secondary question of whether increased genome size ever results in beneficial nucleotypic effects. It's not clear to me that the results in this manuscript do not support the mutation hazard hypothesis since increased body size and increased development time could be mildly deleterious effects, and a decrease in population growth is certainly deleterious. The authors may want to return to this idea in the last paragraph of the Discussion, as something that further work in the system this manuscript established could address.

Minor suggestions below:

Abstract line 29: It seems a little odd to call the population "genetically homogenous" when there are such differences in genome size, even if the core genome is the same across isolates. Maybe "interbreeding population" (redundant, I know) or just "population" or "well-mixed population" as occurs in the Introduction. This is addressed in the Discussion, but the Abstract stands alone.

Introduction line 54: "and vice versa" is not necessary; also, I think it would be more clear to replace "mutational hypotheses" with "this hypothesis."

Introduction line 97: "offspring that is" should be "offspring that are"

Figure 3 line 161: better as "number of males produced in dense clonal cultures"?

Discussion line 168-9: Suggest "that genome size in *B. asplanchnoidis*" (the variation isn't correlated); also "development time" as in the rest of the text.

Discussion line 180: "Nucleotypic control of" seems too strong, suggest "influence on" or "effect on"

Discussion line 203: Suggest comma after "direct effect"

Discussion line 212: More powerful as "Here we provide conclusive"

Figs legend titles. "Influence of" and "Effects of" should probably be the more neutral "Relationship between," as the argument that this is more than correlation comes in the Discussion and is not apparent in the figures themselves.

We would like to thank the four reviewers for their helpful and overall very constructive comments on our manuscript.

We have made substantial changes and additions in the revised manuscript (highlighted in blue in the ms)

Below we reply point-by-point to the comments:

Reviewers' comments in black, *our response in blue and italic*

Reviewer #1 (Remarks to the Author):

In the manuscript “Evidence for nucleotypic effects on the level of a population” the authors take advantage of genome size variation within a population of the rotifer *Brachionus asplanchnoidis* to investigate the effects of genome size on phenotypic correlates, such as cells size and development time. Overall, I really enjoyed reading this paper. I feel it was well thought out and designed and really makes me excited to see what work comes from this. I feel it is one of the few papers to really directly investigate the question they are asking rather than go about it in a roundabout way. I am certainly excited to see this work published, as it definitely will be a good reference for the field of genome size evolution, especially concerning within species genome size change.

I have a few larger comments/questions I feel should be addressed, but most suggestions I have are very minor. Mostly, I wonder how the authors are defining “within population studies”, as some work that is published is within isolines developed from single collection events. How does this differ from what they consider “within population”. Additionally, at least one work was recently published on phenotypic correlates with genome size (although a very different experimental design and less clear results). I wonder what the authors think about how that work relates to their work, or if it does not. There is one area where I hope to get some clarification on statistical methods (Figure 3c). I am not sure if an interaction term was tried? Is there a statistical difference in slopes for morphs when looking at GS vs growth rate.

We had a well-mixed natural population in mind, i.e. a population in which there neither potential nor realized reproductive barriers among individuals. Individuals of such populations are typically outbred, unless the population is extremely small, and they have high genome-wide heterozygosity. We would consider this situation “typical” for most natural populations, including facultative sexual rotifers. This is now stated more explicitly in abstract, introduction, and discussion (lines 27-28, 92, 187). Note that we have also more explicitly referenced a section of the Lynch’s book (Lynch 2007) with the page number (p.34), which was the basic motivation for us to study these questions in such a population (lines 90-92).

Isogenic lines derived from natural (outbred) populations do capture the genetic variation of that population, but there are caveats to be considered due to the low levels of genome-wide heterozygosity in individuals, especially with regard to alleles responsible for genome size variation (discussed in more detail in the next point).

Major:

Lines 84-85: This is where I have my biggest issues/questions about your introduction and discussion. This is an interesting question for sure (impact of genome size on phenotypes).

Certainly, much work has been done on among population differences in genome size. Within a population can be a little trickier, however. How do you consider variation within isolines developed from single populations. For example, the DGRP (Huang et al .2014) have 205 lines of *Drosophila melanogaster* which originated from a single collection event in Raleigh. These isolines could then breed for generations, then be sequenced. There was a large difference in genome size (ranging from 165-190 Mbp or so). Although, phenotypes were not correlated in initially. But a follow-up study (Ellis et al 2014, Plos Genetics) looks at reproductive fitness/phenotypes related to temperature in lines with large and small genome size. Additionally, another recent study published in Genes (Hjelmen et al. 2020) looked at genome size in flies selected for generation time (blow fly) and flies selected for body size (*drosophila melanogaster*). There was variation in either instance, but not clear evidence there was directional change in genome size. Selection for a larger body size did increase variation in genome size, would suggests a reduction in constraint. This study was very different and used pre-existing selection lines (selected for studies of phenotype) and later measured genome size. This may be something worth mentioning or discussing/thinking about.

Thank you for these valuable suggestions. We agree that these studies on genome size variation in isolines warrant a more detailed discussion. We discuss them in lines 242-264:

“There are a few studies that have addressed genome size variation in populations and their phenotypic correlates, but with a slightly different approach. Huang et al. ¹⁰ examined genome size variation in the *Drosophila melanogaster* Genetic Reference Panel (DGRP), a collection of 205 inbred lines that derive from a natural population. Mated females were collected from the field and subjected to 20 generations of full-sib mating in the laboratory. Thus, the DGRP contains a representative sample of naturally segregating variation, but the high inbreeding coefficient of 0.986 predicts that most loci will be homozygous, including the many insertion (+) and deletion (-) alleles across the genome whose net number ultimately determines the genome size of an individual. Thus, we expect fixation of most (-) alleles in strains with the smallest genomes and fixation of most (+) alleles in strains with the largest genomes. Such individuals/genotypes are not very likely to occur in an outbred heterozygous natural population. In a follow up study, Ellis et al. ⁴² selected 50 lines representing the 25 of the largest and 25 smallest genomes from the DGRP, measured life history traits in all 50 of these lines. Thus, they only worked with the most extreme genome sizes. While this among-line genome size variation ultimately stems from structural variants that were already present in the original population, among-line variation in genome size is probably much higher than among-individual variation in a natural population. The DGRP approach offers greater resolution of genome size – phenotype connections than is possible with studies of wild individuals, and indeed this study uncovered some interesting phenotypic correlates ⁴². However, applying these genome size – phenotype relationships to the original population is not straight forward due to the inbred history. In our study, clones represent the natural genotypes and nucleotypes, with essentially the same level of heterozygosity as in the field population. Thus, a correlation between genome size and a phenotypic trait can be directly applied to the population, especially since genome size is inherited like a quantitative trait with a heritability of 1 ¹¹”

We also discuss the Hjelmen et al 2020 study (lines 265-280):

“Another recent study on flies examined whether long-term phenotype selection results in genome size differences between lines that experienced contrasting selection regimes ⁴³. Lines selected for fast/slow development did not change in genome size as expected (i.e., decreasing/increasing in genome size), instead they all converged to the same mean genome size, but with reduced genome size variation compared to the founding populations. Likewise, selection for body size did not result in significant differences among lines in outbred selection lines maintained at large effective population size. However, isolines derived from such populations after 10 generations of full-sib mating experienced greatly increased genome size variation and “bloating” of genome size in several individuals. This observation closely resembles the pattern of increased genome size variation in the DGRP lines (discussed above), and it matches our own observations in B.

asplanchnoidis, where repeated selfing can lead to surprisingly high genome size variation and to increases in genome size compared to the founding individual (see Fig. 3 in ¹¹). Our tentative conclusion is that patterns in genome size derived from highly inbred lines might not be representative for genome size evolution in a natural population. Even though such observations are highly interesting and potentially informative about the basic mechanisms of genome size variation, we decided to not include selfed lines in our present study. ”

Line 124: I may have missed it here, but are these slopes significantly different (between morphs)

We did not test if they are significantly different because the slopes look almost identical.

Figure 1b: I would suggest moving this legend slightly, as it appears to be covering at least one point slightly. Not sure if any are missing behind it. (not likely)

We moved this legend to Fig. 1a, where we could place it without any overlapping with data.

Figure 2a & b: can you specify what the shading indicates? Is this a 95% confidence interval?

Yes, this is the 95% confidence interval. This is now mentioned in the figure legend (line 165)

Figure 2c & d: font size on C and D are far too small to be legible. Can you increase sizes here?

We increased the size of Fig. 2d (actually, now it is called 2c) and we made the former “2c” a separate table (supplementary table S2).

Line 156: What test statistic here for significance? Supplemental data suggests that significant at $p = 0.048$, when combining all data. At least with morph as a component in your model. What does it look like if you don't include morph in your model or if you use an interaction term? It seems that the morph may be driving some. The saccate morph has a negative slope, but the compact does not have a convincing negative slope. Have you run the model with an interaction effect between morph and GS?

*In the revised version we included an interaction between morph and GS (see **Supplementary table S4**), and we think this model is much more appropriate. The interaction is indeed highly significant, indicating that only the saccate morph shows this negative relationship. We mention this in the results (**lines 173-175**) and discussion (**lines 193-197**). In retrospect, it actually makes sense that the large morph is more strongly affected, since its body/egg size is already at the high end of the population distribution.*

Line 170: Again, i am interested to see how much of an effect the morph has, it does really seem different. This could easily be a trick of the eye, though.

See previous point.

Line 172: Again, how would you differ this from the DGRP results in Ellis et al 2014 with temperature/gs reproductive fitness phenotypes? This study had pupation time, pupal mass, eclosion time, etc.

We agree that Ellis et al (2014) provides interesting insights into the relationship between genome size and these phenotypic variables. However, it differs from our study in being based

on a rather extreme set of genotypes, which differ from a well-mixed natural population (see previous point and lines 242-264).

Line 210-211: I do think this would be interesting to see. As referenced in my comments earlier, there does seem to be an effect of genome size and plasticity in development for *Drosophila melanogaster* lines

We will try to address this in a future study.

Minor: I would suggest putting the species name of the organism into the abstract, rather than just “small aquatic metazoan”

Done. See line 33.

Line 22: I'd be careful with a word like "necessary", what about cases for junk dna?

Changed to “expected” see line 23, and also line 49.

Line 52 (references to mutational equilibrium): Some may argue you should include statements that this mutational equilibrium model likely couldn't account for the large differences we see (it's too slow) (Gregory 2004, Insertion--Deletion biases and the evolution of genome size). Another reference that is more recent covering the same idea but at a faster/larger scale would be the "accordion" model in Kapusta et al. 2017. As is, I find this to be a nice summary, though.

Thank you for these valuable suggestions. We now present the mutational equilibrium model in a slightly more nuanced way, and included the references to Gregory 2004 and Kapusta et al. 2017 (lines 55-56)

Line 71: can you give a reference for the statement about “sometimes even genera within families”

We rephrased this sentence slightly and inserted as reference Gregory et al. (2000). See line 76.

Line 114: can you spell out “fl” the first time you use it here.

Done. Line 133.

Reviewer #2 (Remarks to the Author):

This paper poses an interesting general question about variation in genome size, but then the study itself is a bit more specific on interesting biology related to rotifers. The data addresses variation in genome size but also encompasses phenotypically plastic body size morphotypes. My main comment is that the general aspects and justification do not fit well with the specific findings, and so I would recommend that the background to the study is reformulated to fit more closely with the study system and the actual data presented here, which are good.

The main limitations are:

1) It is unclear what the elements are that contribute to variation in genome size. The implication in referring to the nucleotype hypothesis is that it is non-coding, but that evidence is not presented. If it includes any genes, then this could change interpretation of the consequences of the extra material. Then there would be an alternative to the nucleotype hypothesis – that there is a dosage effect of some genes that influence body size – cannot be ruled out by the present data. Hence, results do not support the title. Also, is there any evidence whether they display selfish behaviour or biased transmission?

Actually, we do have data on this! We now mention and cite our preprint on bioRxiv (lines 117-120). These new data are much in line with the notion that “extra DNA in some B. asplanchnoidis individuals is largely non-coding”, which in turn supports our conclusion about nucleotypic effects. Whether these elements display selfish behavior or biased transmission is for sure an interesting question, but it does not affect our study.

2) Claims that this is the first study seem to miss lots of work on accessory B chromosomes. I googled “B chromosomes and body size” and found several papers in mammals and plants reporting either positive or negative correlations of body size with number of B chromosomes, and hence genome size. This work should be referred to and the context for the present study refined accordingly.

Thank you for pointing this out, since this is something that might also occur to other readers.

*It is true that there are considerable number of publications that report correlations between the “number of B-chromosomes” and “phenotypic effects” or “body size”. **However, counts of B-chromosomes cannot be simply taken as a measure of genome size variation.** We discuss this problem and provide the example of maize, where both B-chromosomes and heterochromatic knobs (located on normal chromosomes) are known to contribute to intraspecific genome size variation. Surprisingly, B-chromosomes are inversely(!) correlated to genome size, because heterochromatic knobs overcompensate for the differences due to the B-chromosomes. We conclude that, studies that are solely based on B-chromosome counts cannot be used reliably to make inferences about genome size variation. If one wants to examine the relationship between extra DNA on phenotypes, one has to measure genome size (DNA content) directly. We devoted a separate paragraph in the discussion to this (lines 281-294).*

3) The introduction contains a lot of information about variation in genome size in general, building up to this as a test of the general idea for variation in genome size among eukaryotes. But the discussion does not return to these general questions convincingly – how do the specific results of this study answer the big questions asked at the start? Is this variation really relevant for some of those questions, how does it relate to previous cases of shifts in ploidy or accessory chromosomes or TE contents? Perhaps best would be to rework the beginning and focus more closely on the things of interest here that occupy more time in the discussion, i.e. the rotifer context and possible consequences of variation for them.

We have rephrased several parts of the introduction, so that it is (hopefully) easier to distinguish between the general background and the specific question of our study:

- 1) General background (paragraphs 1&2, lines 40-71): Selection-based hypotheses are one way to explain the genome size diversity across eukaryotes (others are mutation-*

based, and drift-based). Selection-based hypotheses propose that genome size evolves via selection indirectly acting on correlated life history traits. See lines XXX.

- 2) Transition to our specific question (paragraphs 3 & 4, **lines 72-97**). Selection-based hypotheses propose (actually, require) the existence of nucleotypic effects. Only then, selection-based change in genome size can occur on the population level. Nucleotypic effects imply that genome size causally determines phenotypes, and previous evidence for this causal link is controversial. We cite Michael Lynch (**lines 90-92**), quoting from his 2007 book 'The origins of genome architecture': "... the logic underlying the bulk-DNA hypothesis will remain unconvincing until it is demonstrated that (...) heritable within-population variation in genome size significantly covaries with cellular features that are mechanistically associated with individual fitness". **This is the specific question we address in our study**, and we argue that this has so far not been adequately addressed by previous studies.
- 3) The final three paragraphs of the introduction give the specifics of the model organism (Monogonont rotifers, **lines 98-109**; *B. asplanchnoidis*, **lines 110-120**) and study design. We have expanded on biological detail on our model organism, realizing that many readers might be unfamiliar with rotifers and will benefit from this throughout the manuscript. For example, we explain in detail the concept of a 'clone' or 'genotype' in the context of the rotifer life cycle (**lines 103-109**).

By showing that heritable within-population genome size variation significantly covaries with various phenotypic traits associated with individual fitness, our study suggests that selection-based change in genome size could work in principal (**lines 297-299**). Whether it is really common in nature, how often it operates, and under which circumstances, all depends on how easily selection can overcome drift or mutational change (**lines 32-34**). This needs to be addressed in future studies - and we think *B. asplanchnoidis* is a promising system in that respect. (see **lines 299-307**)

4) The word nucleotype is not widely used or known as far as I can tell, and might be better replaced in abstract/title to avoid unnecessary jargon. Neither the title nor abstract mention the study organism - for a single-species experimental study of this kind with lots of biological detail (e.g. morphotypes) should really name the study organism in the title and abstract.

We agree. We replaced 'nucleotype' in title and abstract (**lines 25-27**) accordingly. The title now reads "Linking genome size variation to phenotypic variation within a natural rotifer population". For the sake of brevity, we did not include the species name in the title, but it is now mentioned in the abstract (**line 30**).

Line 51. The distinction between your two mutational hypotheses is not entirely clear as worded. Both of these rely on mutation and drift?

Ultimately, all three hypotheses rely on mutations that change genome size. It is a bit unfortunate and perhaps confusing that the 'mutational hazard hypothesis', which actually emphasizes drift, has mutation (instead of drift) in its name.

The distinction between the 'mutation equilibrium' and the 'mutational hazard hypothesis' should become clearer in our revised version of this paragraph (Key phrases highlighted in **bold**):

“Current hypotheses on genome size evolution in Eukaryotes strongly differ in their emphasis of the evolutionary forces, mutation, selection, and drift. **Theories focusing on mutations** state that the genome size of a species represents a long-term equilibrium of mutations that increase and decrease genome size, by either referring to small indels (^{17,18}, but see ¹⁹), or to the dynamics of transposable elements ²⁰. Variation among taxa is considered the result of **biases in mutational rates**, such that organisms with smaller genome sizes are able to remove DNA at faster rates than organisms with large genome size. In contrast, the 'mutational hazard hypothesis' **prioritizes drift as the main evolutionary force shaping genome size** variation in eukaryotes ^{21,22}. It assumes a constant influx of mutations that increase genome size, which impose a 'mutational hazard' by increasing the genomic target size to deleterious mutations (in particular, harmful gain-of-function mutations). According to this hypothesis, **variation in genome size mainly stems from differences in effective population size (hence, drift)** among taxa ”

See lines 52-63.

Line 73. What is chromatin diminution?

We have added a brief definition in the revised manuscript (Line 79)

Line 88. Are you sure that the different genome sizes do not include gene presence/absence or copy number, which could still permit the dosage explanation?

Actually, we now have pretty good evidence for this. According to our recent study published on bioRxiv, gene density in the regions that contribute to genome size variation in this population is extremely low. Such regions mainly consist of tandemly repeated satellite DNA. Thus, it is rather unlikely that the GS-variable regions contain exactly those genes, in a dosage-dependent manner, that influence phenotypic variation in our study. We mention this in the Introduction (lines 117-120) and in the Discussion (lines 296-298)

Line 96. It would be useful to know more about these elements – are they accessory chromosomes? Do they contain any genes or purely non-coding DNA?

See previous point. We have added information on and a reference to our bioRxiv preprint: “... it has been demonstrated that these independently segregating elements consist of tandemly repeated satellite DNA, with only few interspersed genes or other sequences...” lines 117-120

What explains the saccate and compact morphs? This appears as a complication in your study, which is fine and needs to be addressed, but would help reader to have a bit more explanation of what these are. E.g. are they something like defence polyphenism in *Daphnia*, which people might know about?

*The two morphs were an (unexpected) side discovery (see line 135). In fact, our study seems to be the first to describe this body size dimorphism in *B. asplanchnoidis*. Morphotype doesn't have direct relevance for our study question, so we treat it as an additional random variable. Currently, we can only speculate about the adaptive significance of the two morphs (see discussion, lines 236-240)*

Do the data in figure 1 include artificially selected lines as well or just the natural ones?

They include both types. We have added a reference in the legend of Fig. 1 pointing to table S1, which lists all clones depicted there (line 147).

Actually, we realized that the term “artificially selected” might be a little bit misleading, since these clones are direct (first- or second-generation) outcrossed descendants of the natural clones. Thus, clones with similar genotypes/genome sizes can also be expected in the natural population (so they are actually not that “artificial”). We thus deleted the instances of ‘artificially selected’ in this paragraph and instead provide a more accurate description of this set of clones (lines 124-125):

“To this end, we analyzed body size and egg size variation in 141 genotypes of the OHJ-population, which were either sampled directly as resting eggs or their outcrossed offspring.”

Line 137. Not fully explained what direct and indirect effect you are talking about – please clarify what factors these effects are between and via in each case.

Changed to (lines 156-157):

“...to distinguish between a direct effect of genome size on EDT and an indirect effect (i.e., an effect of genome size on EDT via egg size)”

Line 140-143. Losing focus here, why are you interested in the male eggs and their hatching time? Not clear to the reader. Could this be tied to differences in genome size as well if males are haploid?

The (longer) development times of males are actually quite interesting since they show that other factors (in this case, sex) can influence EDT. If we would only consider genome size, we would expect that males, which are haploid, have faster development than (diploid) female eggs. This is obviously not the case. We now discuss this briefly and suggest that sex-specific differences in embryonic development may be responsible. Within male eggs, we would predict a similar increasing relationship between genome size and EDT as in female eggs See lines 221-229.

Line 156. “Population growth rate declined significantly with genome size” – from figure 3 it looks like this is just for the saccate morph. Previous parts separated effect between the two morphs it is unclear why you lumped everything together for this part.

*Correct – this is just in the saccate morph. Reviewer 1 made the same point. In the revised manuscript we included an interaction term in the model (see **Supplementary table S4**). And we provide an explanation in the discussion why this may just affect the saccate morph. See results (lines 174-176) and discussion (lines 193-197)*

Line 225. Please can you add the sampling date.

We added “in autumn 2011” (the exact day does not matter, since we sampled resting eggs). Line 309.

With permission of the editor, my research group helped to review this manuscript, but the final report is my own.

Reviewer #3 (Remarks to the Author):

This manuscript details a study that identifies the existence of nucleotypic effects within a genetically homogeneous population. The authors generally conclude that the body size, egg size, and embryo development time can each contribute to the genome size. While this study demonstrates some new approaches, I possess several significant reservations regarding the

study that preclude its publication in Commun. Biol. below. Some points of concern with the study are outlined below, alongside some more specific line-by-line comments. Given my concerns, I unfortunately must recommend rejection of the manuscript in its present form.

Major comments

(1) Novelty: It is not clear what the novelty of this study is. Although the authors said that the two morphotype (saccate and compact) outweigh the nuclear effects, they have not presented the relevant mechanisms. The abstract describes evidence for additional DNA and nucleotype effects, but the authors only provide correlations between the genome and egg size, embryo development time and phenotypes. Furthermore, two other studies are given (refs 24 and 25) that already looked at the correlation between genomes size and life history traits. Additionally, there is already substantial evidence of a weak relationship between genome size variations and organic complexity (C-value enigma). Indeed, the crucial considerations of this paradox were mentioned in the Tomas 1971 review and in Nature Ecology & Evolution by Liedtke. Therefore, it is unclear how this current research has added any novel or improved insight.

The above cited examples for “evidence of a weak relationship between genome size variations and organic complexity” essentially involve correlations above the species/population level. We (and others) argue that these broad-scale correlations cannot be taken as demonstrations of nucleotypic effects on the within-population level. See lines 75-96.

The novelty of our study consists in addressing these phenotypic correlations in a well-mixed natural population that shows heritable variation in genome size (lines 37-38, 87-88). This is important, because selection-based change in genome size is referring to exactly this situation. See lines 90-94.

Related to this, please note that the reviewers 1 & 2 made suggestions on earlier studies that are closely related to our work (DGRP lines, B chromosomes), and we discuss these now and explain in which way our study is different (lines 242-294).

(2) The Introduction needs more information on the species' biology. If, as the authors said, resting eggs from each clone are unique genotypes, why would you expect correlation between genetic variation and life history in cyclic parthenogenesis? (The authors cited the need to avoid biases in the genetic backgrounds among population samples) (L89-90). Therefore, I would argue that cyclic parthenogenesis is not appropriate because too many variables must be taken into account to compare genomic variation with life history (i.e., mutation, meiosis).

We have added a new paragraph to the introduction in which we provide more background on monogonont rotifers (lines 98-109) and on this species (lines 110-120). This also explains in more detail the concept of ‘clone’ and ‘genotype’ in the context of the rotifer life cycle. Cyclical parthenogenesis, with its possibility to propagate sexually recombined genotypes clonally, is actually extremely useful for our study, in terms of replication and measuring multiple different traits in the exact same genotype.

(3) Unfortunately, the authors did not examine the single nucleotide polymorphism to estimate genotypic diversity among their samples in population-level analysis. In addition, the morphotype is more significant than the nucleotypic effect (L197-198), but it is not clear because it is not specifically mentioned. Without a fully-factorial design, the authors cannot

really distinguish the correlation between genome size and life history characteristics, and this precludes many of their conclusions.

We would like to point out that we did use Amplified Fragment Length Polymorphisms (which is also a marker that can measure genotypic diversity) in a previous study on the same population (this is mentioned in lines 188-191). This previous study has shown that there is no genetic structure within the population (even though this population was highly differentiated from geographically separated populations of the same species), which supports our assumption that this is a well-mixed population with no reproductive barriers. This enables us to draw meaningful conclusions about the microevolutionary consequences from the observed genome size – phenotype correlations.

(4) The description of the Methods requires more detail in order of the reader to gauge the methodological approach. There were many instances where I could not understand what the authors were trying to say. I had a hard time understanding the experimental procedure, and am still not sure what was done.

It is a little bit difficult for us to address this point without a few concrete examples on which parts were particularly difficult to understand...

*We would like to point out that actually much detail on the measurements on body size, egg size and embryonic development time was given in the **Supplementary Information** (see **SI-lines 22-97**). We did not include this information in the methods of the main paper, because we thought it was too technical, and after all, we “only” measured body/egg size and developmental times.*

(If this is explicitly wanted, we would be willing move all of the detailed description of the image analysis system to the methods of the main paper)

(5) The Discussion section lacks of an integrative analysis of its three components (nucleotypic effect, life history traits, and morphotype). In Result, the authors stated that the correlation between genome size, morphotype, and life history traits, but in-depth analysis of their relationship in Discussion was absent. At least, an effort should be made in this direction in the Discussion of results. Thus, this section should be rewritten, as it lacks depth.

Our revised and greatly extended discussion should provide this additional depth. Following the advice of reviewer 1, we now discuss in detail the relationship between our study design and those on the DGRP lines and the Hjelman et al study. We also discuss our study in relation to studies on B-chromosomes (reviewer 2).

(6) Further, despite using statistical figures such as PCA and a generalized linear modelbased calculation, there are no details provided in this manuscript for how the calculation was performed or of the tool that was implemented. Therefore, the statistics section needs to provide substantially more detail as, in this current form, there is not enough information from the statistical design to adequately judge the statistical approach.

We used R and add-on packages for all statistical analyses. We have now added these details and listed all R-packages. See lines 356-373.

Specific comments

L34: The abstract should include a brief statement explaining the primary results and a short concluding sentence. The section on 'selection' should be omitted as this part of the study was not investigated.

*The primary results are reported in **lines 26-32**.*

Based on what we have seen in other recent articles in 'Communications Biology', many abstracts end with a brief conclusion about the immediate implications for the field. Since we have shown that (heritable) within-population variation in genome size affects the phenotype, we believe that the following outlook for future studies is appropriate (lines 32-34:

“Our results suggest that selection-based change of genome size can theoretically operate in this population, provided it is strong enough to overcome drift or mutational change of genome size.”

L83-87: This is not clear to me. In this experiment, isn't an asexual rotifer more suitable to avoid total disruption due to genetic differences? (Line 241: you used different clones in this experiment).

We are sorry to hear that this point did not come across. Our basic experimental units are the 141 genotypes deriving from the natural OHJ-population, each of which was genetically unique (because it hatched from a sexually produced resting egg). Furthermore, they differed in genome size. We cultured these genotypes clonally, but the latter was just for replication and for measuring different traits in the same genotype.

If instead we would look at just “an asexual rotifer” (one clone??) there would be no genome size variation at all to begin with. Also, we don't think that sexual reproduction in a well-mixed natural population leads to “total disruption due to genetic differences”.

*We hope that the now more explicit citation of Lynch 2007 clarifies that nucleotypic effects should best be studied on the level of a population (**lines 90-92**)*

L115-116: What are the criteria for selecting the two types? Many studies have recognized three major morphotypes: large (L), small-medium (SM), and small (SS). (Fu et al., 1991).

*There might be a misunderstanding here: The morphotypes in our study are based on PCA-classification of body shape (**lines 358-366**). This is an example of discrete morphological variation within a species. By contrast, the three “morphotypes” in (Fu et al. 1991 {1991}) date back to a time when the so called “*B. plicatilis* species complex” was still considered a single species. However, nowadays it is widely established that the *B. plicatilis* complex consists of at least 15 different species - and our study species *B. asplanchnoidis* is one of them (Mills, Alcantara-Rodriguez et al. 2017). Thus, there is no resemblance at all between the morphotypes in Fu et al. (which should, in retrospect, better be called “clades” because each encompasses many species) and the ‘saccate’ and ‘compact’ morphotype in our study, which are true morphological variants within a species.*

L171-172: What if different populations had different trends?

*Our study focusses on a general mechanism and we investigate it in one specific animal population. There are many such ‘case studies’ in the literature, e.g., on Darwin finches, St. Kilda Soay Sheep, *Drosophila* DGRP lines. Neither our study nor any of the others can exclude the possibility that some other population in the world might yield somehow different results.*

L175-179: I cannot verify the validity of this statement based on the data presented in the

figures. Please point the reader to the data identifying the inheritance of body size in individual clones.

*We think the notion that “body size is under polygenic control in most animals” is widely established. We clarified this by inserting a citation of “Introduction to Quantitative Genetics” (Falconer and Mackay 1996) right after mentioning this, and by slightly rephrasing the sentence. See **lines 218-220**.*

(P.S.: in our Edition of Falconers book, several examples of heritability of body size/weight are listed in Tab. 10.1)

L192-193: Include a reference for the statement that non-genetic polymorphisms can occur in several rotifer species.

*We moved the reference of Gilbert (2017) one sentence earlier. **Line 233**.*

L199-201: I am confused by this section. Does it mean that the nucleotypic effect is more important than the morphotype because the selection works well on all the phenotypes or the phenotypes are sometimes observed in only one form? This statement is perplexing.

We removed these two sentences.

Reviewer #4 (Remarks to the Author):

The authors take advantage of a species, the monogonont rotifer *Brachionus asplanchnoides*, that has reproductively compatible isolates of drastically different genome size to look for “nucleotypic effects” – phenotypic effects of genome size independent of genetic information. They convincingly demonstrate a positive correlation between genome size and female adult body size, egg size, and embryo development time. This is the most direct evidence yet of nucleotypic effect, free of the confounding variables of evolutionary and ecological divergence that comes with cross-species comparisons, and suggests that *B. asplanchnoides* would make a suitable system to study whether these phenotypes are under selection. The paper is well written and admirably concise, and I recommend that it be accepted after minor revision.

The authors definition of nucleotypic effects seems to assume that the “extra” DNA segregating in the population is not transcriptionally active (“non-coding” “independent of information content”). It is probably full of transposons, and if they have recently been introduced into a new genomic environment, they may be quite active and conferring their own deleterious effects. If the authors could say anything about what is known about the nature of the excess DNA it would be helpful.

*We have recently published relevant results on bioRxiv, showing that the genomic regions that contribute to genome size variation in this population consist mostly of tandemly repeated satellite DNA (Stelzer, Blommaert et al. 2021). We mention this now in the introduction (**lines 117-120**) and in the discussion (**lines 296-298**). Overall, this new study greatly supports our assumption about the low information content of extra DNA in *B. asplanchnoides*.*

The authors describe two morphotypes, one considerably larger than the other. This is an interesting thing in and of itself for a different publication, but the fact that the two sizes have nearly the same relationship between genome size and both body size and egg size is a powerful indicator of how these nucleotypic effects scale. It would be interesting if these relationships held with males. Fig S5 briefly touches on development time of male embryos—

male embryos develop faster than females from the same clone—but a more interesting question is whether males show the same relationships as females in Figs 1 and 2.

We mentioned the male embryonic development times, because they suggest that sex-specific differences might be an additional variance component of EDT. This is now addressed in the Discussion (lines 221-229).

We agree it could be hypothesized that male eggs show the same increasing relationship of EDT with genome size as (asexual) female eggs (see line 226-227). However, this would be challenging to measure: males often differ in genome size, even if they were produced by the same clone, due to independent segregation of the elements that contribute to genome size variation in this species (see Stelzer, Pichler et al. 2019). Thus, we would have to measure EDT in individual male eggs, and then measure the genome size of the male that hatched from the egg. This is currently not possible with flow cytometry, since an individual male rotifer does not contain enough cells for even one measurement.

I don't understand Figure 3c. The authors state that population growth rates were significantly reduced at higher genome sizes and present the results of a GLM in Supplemental, but did this model lump compact and saccate morphs? Eyeballing Fig3c the two seem to be different, with saccate decreasing with genome size and compact remaining constant. If population growth rates are really reduced with increasing genome size this would seem to be a serious deleterious effect.

*You are right, the slope of the saccate morph is indeed steeper (this was also mentioned by the other reviewers). Our previous GLM did not account for this since it did not include an interaction term, which was clearly inappropriate. Thus, we have now included an interaction “morphotype x genomesize” to the model (see **Supplementary table S4**), and this is indeed highly significant, showing that the decrease in population growth at large genome size only applies to the saccate morph. We mention this in the results (lines 173-175) and discussion (lines 193-197).*

In the Abstract, the authors state that the observed nucleotypic effects “account for” difference in genome size. This seems to conflate two ideas: one, that changes in genome size have non-genic phenotypes, and two, that these phenotypes can be subject to selection in a meaningful way. The results support the first idea but as the authors make clear in the discussion more work is needed to establish the second. Perhaps something like “Nucleotypic effects are predicted to arise from variation in genome size—they refer to...” would be better.

Thank you for pointing to this lack of precision in our wording. In the revised version, we do not use “nucleotypic effects” any more in the abstract (also in response to a suggestion of reviewer 2), but instead use only its definition: “phenotypic effects of DNA that are independent of its DNA content”, which should avoid any potential ambiguity. See line 26.

In a similar vein, the organization of the second paragraph of the introduction implies that nucleotypic effects are an aspect of selection-based hypotheses for variation in genome size. I'm not sure this has to be true, or that the “mutation hazard” and “selection based” hypotheses are mutually exclusive. The first assumes excess DNA is mildly deleterious and the second assumes it can be mildly beneficial or mildly deleterious. The first emphasizes the importance of drift vs selection in small populations while the second assumes extremely efficient selection over a short period of time. The question is whether the selective effects are strong enough (or populations large enough) to overcome drift, with the secondary question of whether increased genome size ever results in beneficial nucleotypic effects. It's not clear to

me that the results in this manuscript do not support the mutation hazard hypothesis since increased body size and increased development time could be mildly deleterious effects, and a decrease in population growth is certainly deleterious. The authors may want to return to this idea in the last paragraph of the Discussion, as something that further work in the system this manuscript established could address.

*We would like to clarify that the purpose of the second paragraph (lines 52-71) was to introduce the reader to the general theoretical background of genome size evolution by presenting the three main hypotheses of the field. This is the background – our study does neither test these hypotheses nor evaluate them against each other. Rather, we focus on an important **premise** of selection-based hypotheses: nucleotypic effects (this is developed in the next two paragraphs, lines 72-97).*

And it is exactly this premise that has been questioned on the basis of the empirical evidence (correlations above the species-level), specifically by Michael Lynch, quoting page 34 from his 2007 influential book ‘The origins of genome architecture’:

“... the logic underlying the bulk-DNA hypothesis will remain unconvincing until it is demonstrated that (...) heritable within-population in genome size significantly covaries with cellular features that are mechanistically associated with individual fitness”

This was the main motivation for our study. We found out that this premise is indeed fulfilled in the OHJ-population, which “rehabilitates” selection-based theories from the above-mentioned criticism.

*Nevertheless, we do not at all consider our study “proof” for selection-based theories, rather we suggest that selection-based change in genome size could work in this population (lines 298-300 and lines 32-34). Whether selection-based change in genome size actually is common in nature, how often, and under which circumstances it can become dominant, indeed depends on how easily selection can overcome drift. However, this needs to be addressed in future studies - and we think *B. asplachnoidis* is a promising system in that respect. This is stated in the last paragraph of the discussion (lines 303-307).*

It might be tempting, but we do not think it would be appropriate, to favor one of the three hypotheses based on the presumed beneficial or deleterious effects inferred in our study. Specifically, we would not want to go as far to interpret any of the trait changes in our study are unequivocally ‘beneficial’ or ‘deleterious’. For instance, whether large body size is advantageous, or not, likely depends on specific factors of the environment (resource levels, predators, etc.). Likewise, a slower development may not be deleterious in all situations: even though offspring from larger eggs may take longer to develop, which will lower its growth, it may also be more starvation resistant and “protected” for a longer time in a developmental stage that does not need to feed. Even the decrease in population growth at the large morph/large GS might not be unconditionally deleterious, since it was measured in a rather benign laboratory environment. There could be other environmental circumstances (starvation periods, low temperature, etc.) where the large morph/large GS clones are in fact superior.

Minor suggestions below:

Abstract line 29: It seems a little odd to call the population “genetically homogenous” when there are such differences in genome size, even if the core genome is the same across isolates.

Maybe “interbreeding population” (redundant, I know) or just “population” or “well-mixed population” as occurs in the Introduction. This is addressed in the Discussion, but the Abstract stands alone.

We fully agree. We now call it a ‘well-mixed natural population’. Line 31.

Introduction line 54: “and vice versa” is not necessary; also, I think it would be more clear to replace “mutational hypotheses” with “this hypothesis.”

Changed accordingly.

Introduction line 97: “offspring that is” should be “offspring that are”

Changed accordingly.

Figure 3 line 161: better as “number of males produced in dense clonal cultures”?

Changed accordingly.

Discussion line 168-9: Suggest “that genome size in *B. asplanchnoidis*” (the variation isn’t correlated); also “development time” as in the rest of the text.

Changed accordingly.

Discussion line 180: “Nucleotypic control of” seems too strong, suggest “influence on” or “effect on”

Changed to: nucleotypic effects on body size are...

Discussion line 203: Suggest comma after “direct effect”

Done.

Discussion line 212: More powerful as “Here we provide conclusive”

Thanks. Changed accordingly.

Figs legend titles. “Influence of” and “Effects of” should probably be the more neutral “Relationship between,” as the argument that this is more than correlation comes in the Discussion and is not apparent in the figures themselves.

We fully agree. Changed as suggested.

References

*Ellis, L. L., W. Huang, A. M. Quinn, A. Ahuja, B. Alfrejd, F. E. Gomez, C. E. Hjelmén, K. L. Moore, T. F. C. Mackay, J. S. Johnston and A. M. Tarone (2014). "Intrapopulation genome size variation in *D. melanogaster* reflects life history variation and plasticity." *PLoS Genetics* 10(7): e1004522.*

Falconer, D. S. and T. F. C. Mackay (1996). Introduction to quantitative genetics, Pearson.

*Fu, Y. K., K. Hirayama and Y. Natsukari (1991). "Genetic divergence between S and L type strains of the rotifer *Brachionus plicatilis* O.F. Müller." *Journal of Experimental Marine Biology and Ecology* 151: 43-56.*

*Gilbert, J. J. (2017). "Non-genetic polymorphisms in rotifers: environmental and endogenous controls, development, and features for predictable or unpredictable environments." *Biological Reviews* 92(2): 964-992.*

*Gregory, T. R., P. D. Hebert and J. Kolasa (2000). "Evolutionary implications of the relationship between genome size and body size in flatworms and copepods." *Heredity* 84: 201-208.*

Lynch, M. (2007). The origins of genome architecture. Sunderland, MA, Sinauer.

Mills, S., J. A. Alcantara-Rodriguez, J. Ciro-Perez, A. Gomez, A. Hagiwara, K. H. Galindo, C. D. Jersabek, R. Malekzadeh-Viayeh, F. Leasi, J. S. Lee, D. B. M. Welch, S. Papakostas, S. Riss, H. Segers, M. Serra, R. Shiel, R. Smolak, T. W. Snell, C. P. Stelzer, C. Q. Tang, R. L. Wallace, D. Fontaneto and E. J. Walsh (2017). "Fifteen species in one: deciphering the *Brachionus plicatilis* species complex (Rotifera, Monogononta) through DNA taxonomy." *Hydrobiologia* **796**(1): 39-58.

Stelzer, C.-P., J. Blommaert, A.-M. Waldvogel, M. Pichler, B. Hecox-Lea and D. B. Mark Welch (2021). "Genome structure of *Brachionus asplanchnoidis*, a Eukaryote with intrapopulation variation in genome size." *bioRxiv*: 2021.2003.2009.434534.

Stelzer, C. P., M. Pichler, P. Stadler, A. Hatheuer and S. Riss (2019). "Within-Population Genome Size Variation is Mediated by Multiple Genomic Elements That Segregate Independently during Meiosis." *Genome Biology and Evolution* **11**(12): 3424-3435.

REVIEWERS' COMMENTS:

Reviewer #1 (Remarks to the Author):

In this manuscript, the authors investigate phenotypical correlates of genome sizes in a natural population of rotifer. The authors find heritable variation in genome size and that genome size is positively correlated to body size, egg size, and embryonic development time. I feel the manuscript is well written and clear. This is my second time reviewing this manuscript and I feel the authors did a nice job addressing reviews from all reviewers. I only have a couple minor suggestions/comments.

Line 25: "selection indirectly acting on correlated life history traits". This statement sounds like selection is acting indirectly on the phenotypic correlates and possibly directly on genome size. I would assume that the intention is that indirect selection occurs on genome size due to selection on the phenotypic correlates. Can this be clarified?

Line 133: Thank you for spelling out femtoliters, but can you add "fl" in parentheses afterwards?

Figure 2C: maybe put "Genome Size" on two lines so the font can be larger? It seems a little strange to see three different font sizes on the plot, unless the font size signifies something, which isn't apparent.

Nice work

Reviewer #3 (Remarks to the Author):

The authors answered my tricky questions well and explained the purpose of this paper accurately.

Reviewer #4 (Remarks to the Author):

I commend the authors on their thoughtful and thorough response to reviewers. Their responses and the revisions to the manuscript have addressed all of my concerns. The extended discussion of the saccate form and males strengthens the manuscript and the review of fly data places the current study in context. I recommend acceptance of the manuscript. Below are a few editorial suggestions that the authors can take or not and that do not require another round of review.

1. Abstract: I don't think the last sentence is an improvement over the previous version. Isn't the statement "selection can theoretically operate provided it is strong enough to overcome drift and mutation" always correct? Are you really saying anything?

2. Introduction, lines 52-53: More natural to say "Current hypotheses of genome size... differ in their emphasis on the evolutionary forces of mutation, selection, and drift."

3. Discussion, line 194: no comma between "size" and "might"

4. Discussion, lines 195-195: More natural to say "However, these measurements were done in a ... there could be situations where"

5. Discussion, line 221: I read "Sex is another source" as talking about mixis. Maybe "Sexual dimorphism is another source"?

6. Discussion, line 227: I suggest "This could be addressed" or more succinctly, "An examination of this possibility would require..."

We would like to thank the reviewers for their positive evaluation and for their final comments.

Below we reply point-by-point to the remaining comments:
Reviewers' comments in black, *our response in blue and italic*

REVIEWERS' COMMENTS:

Reviewer #1 (Remarks to the Author):

In this manuscript, the authors investigate phenotypical correlates of genome sizes in a natural population of rotifer. The authors find heritable variation in genome size and that genome size is positively correlated to body size, egg size, and embryonic development time. I feel the manuscript is well written and clear. This is my second time reviewing this manuscript and I feel the authors did a nice job addressing reviews from all reviewers. I only have a couple minor suggestions/comments.

Line 25: "selection indirectly acting on correlated life history traits". This statement sounds like selection is acting indirectly on the phenotypic correlates and possibly directly on genome size. I would assume that the intention is that indirect selection occurs on genome size due to selection on the phenotypic correlates. Can this be clarified?

*Thank you for pointing on this potential ambiguity. This sentence should be less ambiguous if we simply remove the word 'indirectly' → selection acting on correlated life history traits (see **Line25**)*

Line 133: Thank you for spelling out femtoliters, but can you add "fl" in parentheses afterwards?

Done. (Line 134)

Figure 2C: maybe put "Genome Size" on two lines so the font can be larger? It seems a little strange to see three different font sizes on the plot, unless the font size signifies something, which isn't apparent.

*We have put "Genome size" on two lines. The font size is now larger. (see **Fig 2c**)*

Nice work

Reviewer #3 (Remarks to the Author):

The authors answered my tricky questions well and explained the purpose of this paper accurately.

Reviewer #4 (Remarks to the Author):

I commend the authors on their thoughtful and thorough response to reviewers. Their responses and the revisions to the manuscript have addressed all of my concerns. The extended discussion of the saccate form and males strengthens the manuscript and the review of fly data places the current study in context. I recommend acceptance of the manuscript. Below are a few editorial suggestions that the authors can take or not and that do not require another round of review.

1. Abstract: I don't think the last sentence is an improvement over the previous version. Isn't the statement "selection can theoretically operate provided it is strong enough to overcome drift and mutation" always correct? Are you really saying anything?

*We can understand that from a theoretical and general standpoint, this may sound a little bit like a truism. However, testing whether exactly this precondition is fulfilled, will be a crucial empirical question in future studies, to distinguish among different theories about genome size evolution. So, we removed the "theoretically", but otherwise kept the sentence the same. (see **Line 33**)*

2. Introduction, lines 52-53: More natural to say "Current hypotheses of genome size... differ in their emphasis on the evolutionary forces of mutation, selection, and drift."

*Changed as suggested. **Line 52-53***

3. Discussion, line 194: no comma between "size" and "might"

Comma deleted.

4. Discussion, lines 195-195: More natural to say "However, these measurements were done in a ... there could be situations where"

*Changed as suggested. (**Lines 201-203**)*

5. Discussion, line 221: I read "Sex is another source" as talking about mixis. Maybe "Sexual dimorphism is another source"?

*Changed as suggested. **Line 227.***

6. Discussion, line 227: I suggest "This could be addressed" or more succinctly, "An examination of this possibility would require..."

*Thanks. We chose your second suggestion. **Line 233.***